# Passerini-type reaction of boronic acids enables $\alpha$-hydroxyketones synthesis

Kai Yang[1], Feng Zhang[1], Tongchang Fang[1], Chaokun Li[1], Wangyang Li[1] & Qiuling Song [1]✉

Multicomponent reactions (MCRs) facilitate the rapid and diverse construction of molecular scaffolds with modularity and step economy. In this work, engagement of boronic acids as carbon nucleophiles culminates in a Passerini-type three-component coupling reaction towards the synthesis of an expanded inventory of α-hydroxyketones with skeletal diversity. In addition to the appealing features of MCRs, this protocol portrays good functional group tolerance, broad substrate scope under mild conditions and operational simplicity. The utility of this chemistry is further demonstrated by amenable modifications of bioactive products and pharmaceuticals as well as in the functionalization of products to useful compounds.

[1] Key Laboratory of Molecule Synthesis and Function Discovery, Fujian Province University, College of Chemistry at Fuzhou University, Fuzhou, Fujian 350108, China. ✉email: qsong@hqu.edu.cn

α-Hydroxyketones (also known as acyloins) are structural units ubiquitously found in natural products[1–5] and pharmaceuticals[6,7]. They are also oft-employed synthetic precursors in a panel of high-value transformations (Fig. 1a)[8–13]. The construction of these important molecules is therefore the subject of substantial synthetic efforts[14]. Traditional benzoin condensation method assembles α-hydroxyketones via condensation of different aldehydes, thus limits its applicability within this substrate class[15–17]. The alternate oxidative pathways that encompass α-hydroxylation of ketones[18–22] and ketohydroxylation of olefins[23–27] are certainly enabling, but continues to be challenged in terms of substrate diversity and poor selectivity. Hence, devising complementary routes towards these useful entities from readily available starting materials is highly relevant and desirable.

Multicomponent reactions (MCRs) are often prized for their concise and modular features in forging complex molecules with synthetic and biological interest[28–36]. The representative Passerini reaction[37–46] or Ugi reaction[47–54] efficiently assembles α-acyloxyamides or α-acylaminoamides from several reactant components via the intermediacy of nitrilium species in single-pot operation (Fig. 1b)[55–58]. Interception of this electrophilic intermediate in Passerini reaction pathway by carbon nucleophiles (in place of conventionally used carboxylic acids) would offer an intriguing access to α-hydroxyketone products; yet such synthetic maneuver remains underexplored[59,60]. Central to the successful establishment of this chemistry would lie in choosing suitable carbon nucleophiles that would not interfere with the formation of the nitrilium intermediate while possess sufficient nucleophilicity to capture this electrophile.

On the other hand, boronic acids are easily available, benign and common building blocks for C-C bond cross-coupling reactions, in both transition-metal catalysis[61,62] and metal-free catalysis regimes[63–75]. In boronic acid-Mannich reaction (or Petasis reaction), for instance, the nucleophilic feature of boronic acids effects the formation of boron "ate" complex, leading to functionalized amines following 1,3-metallate migration (Fig. 1c)[76–80]. To this end, a recent endeavor of our group has unraveled a 1,4-metallate shift of boron "ate" nitrilium species generated from nitrile oxide and arylboronic acid, thus mediating stereospecific formation of C-C bond between oxime chlorides and arylboronic acids under metal-free conditions[81]. Grounded in these knowledges, we envisioned that a boron "ate" nitrilium intermediate could be released from co-treatment of aldehyde, isocyanide, and boronic acid; 1,4-metallate shift of which will invoke C-C bond coupling and α-hydroxyketones could be revealed on hydrolysis (Fig. 1d). Here, we disclose the development of a Passerini-type coupling reaction, which afforded α-hydroxyketones from the combination of readily

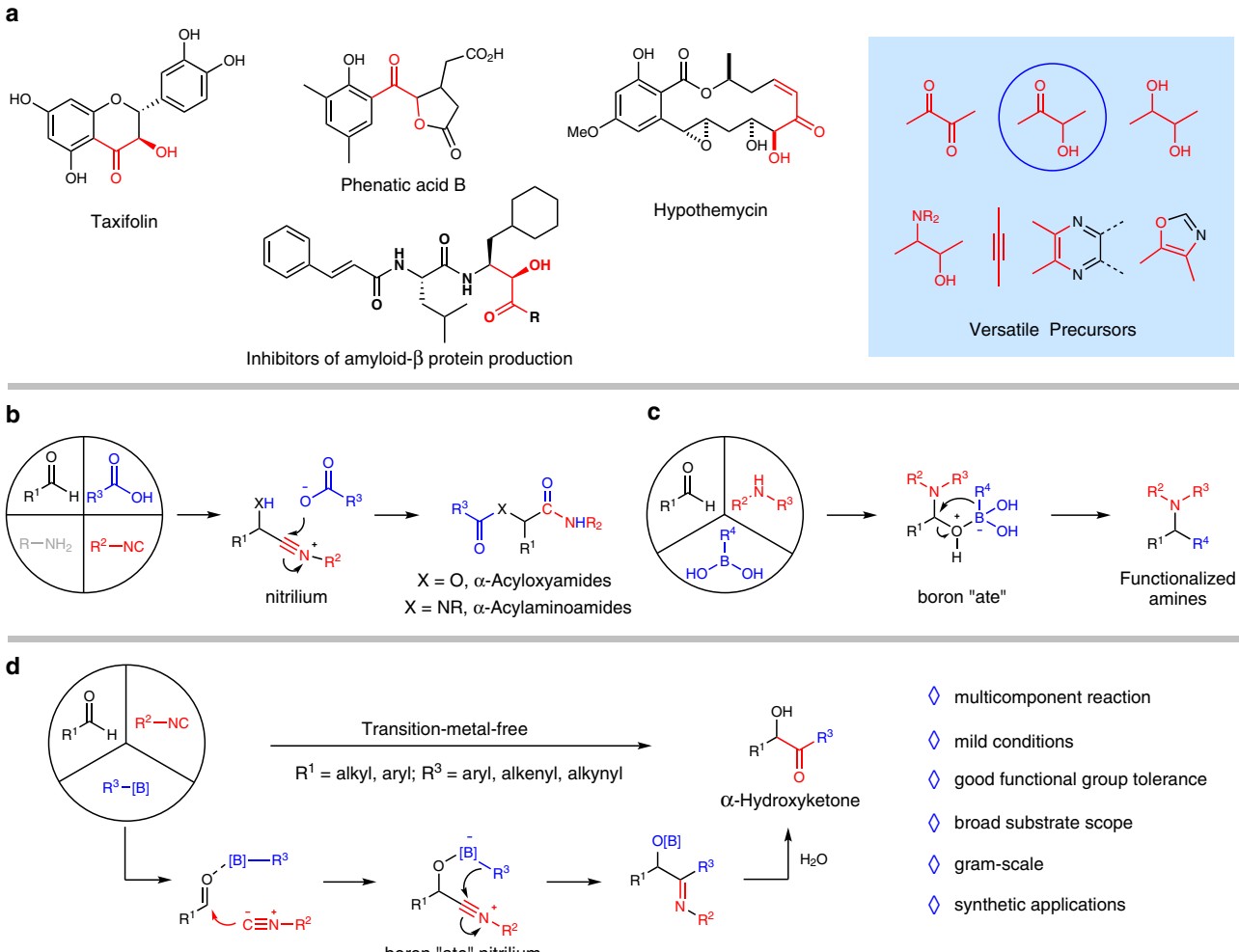

**Fig. 1 Precedent works and proposed Passerini-type coupling reaction with boronic acids as nucleophilic agents. a** α-Hydroxyketones in bioactive moleculars or as synthetic precursors. **b** Classic Passerini or Ugi reaction. **c** Petasis boronic acid-Mannich reaction. **d** Passerini-type coupling reaction of boronic acids (this work).

available aldehydes, isocyanides, and boronic acids (aryl, alkenylboronic acids, and alkynyl trifluoroborate salts) under transition-metal-free conditions. Mild reaction conditions, ease of execution, high functional group tolerance, broad substrate scope, and utility are practical features of this methodology.

## Results

**Investigation of reaction conditions**. Exploratory investigations towards our envisioned Passerini-type reaction involving boronic acids were conducted with phenylpropyl aldehyde (**1a**), tertbutyl isocyanide (**2a**) and 4-methoxyphenyl boronic acid (**3a**) as test substrates (Table 1). To our delights, simple mixing of the three reactants (**1a**, **2a**, and **3a**) without any other additive in DCM furnished the desired α-hydroxyketone product **4a** in 60% isolated yield (entry 1). A solvent screen of DCE, MeCN, toluene, MeOH, and THF revealed that the best reaction efficiency was endowed by $CHCl_3$, whereas using MeOH caused a complete reaction inhibition (entries 1−7). As reaction temperature was decreased to 10 °C, the yield of **4a** improved to 68% (entry 8). Binary mixture of $CHCl_3$ and water in a ratio of 7:3 (entries 9−11) minimally but meaningfully enhanced the delivery of **4a** to 72% yield (entry 11). This has guided our subsequent study of mixed solvent system with $CHCl_3$ against various buffer solutions (entries 12−15) where the combination with pH = 8.0 buffer delightfully provided 81% yield of target product (entry 14). We reasoned that a basic reaction medium could sequester the byproduct $B(OH)_3$ generated during reaction, thus promoting this boronic acid-involved Passerini-type reaction. It was further established that on replacement of tertbutyl isocyanide (**2a**) with cyclohexyl isocyanide (**2b**), benzyl isocyanide (**2c**), or ethyl 2-isocyanoacetate (**2d**), formation efficiency of α-hydroxyketone product **4a** was diminished (entry 16). None of the other ratios of the three reagents resulted in higher yields (entries 17−18).

**Scope of aldehydes**. Having optimized the model coupling of this Passerini-type reaction, we examined the generality of these conditions with respect to a range of aldehyde components (Fig. 2). Delightfully, diverse aliphatic aldehydes were aptly transformed in moderate to high yields. Phenylpropyl aldehydes with strong electron-withdrawing groups and 3-(furan-2-yl)propanal furnished the α-hydroxyketone products **4b**–**4d** in 66% to 90% yields. The chain length of aldehydes posed no effect on the effectiveness of this reaction, providing respective α-hydroxyketones (**4e**–**4g**) in moderate yields. Primary aldehydes bearing ester, adamantyl, and benzyloxy moieties were tolerated well to yield **4h**–**4j** in moderate efficiencies. Secondary aldehydes comprised of acyclic and cyclic analogs (cyclopropyl, cyclohexyl, piperidinyl) were incorporated in **4k**–**4q** with moderate to good yields as well. The diastereomeric ratios (dr) of compounds **4l** and **4n** are 1.13:1 and 1.38:1. Comparable outcome was observed for a tertiary 1-phenylcyclobutane-1-carbaldehyde substrate, which afforded **4r** in 54% yield. It merits mention that transformation of paraformaldehyde has given rise to **4s**, which serves as versatile synthetic intermediate for a variety of bioactive molecules. More importantly, this reaction was well suited to diverse aromatic aldehydes when treated in concert with cyclohexyl isocyanide (**2b**). The electronic property and the position of substituents on the benzene ring had minimal bearing on the efficiency of this transformation. Neutral (**4t**), electron-rich (**4u**–**4y**), or electron-deficient (**4z**–**4aa**) functionalities found good compatibility and were left unscathed in respective molecular outputs. The accommodation of halogen substituents (**4ab**–**4ae**) signified potential structural elaborations from these handles. Fused ring reactants including 2-naphthaldehyde (**4af**) and 1-naphthaldehyde (**4ag**) were also suitable candidates for this MCR.

**Scope of boronic acids**. This protocol featured an admirable scope with respect to arylboronic acid substrates (Fig. 3). For electron-rich

**Table 1 Optimization of the reaction conditions[a].**

| Entries | Isocyanide | 1a:2:3a | Solvent | Temp. (°C) | Yield (%)[b] |
|---|---|---|---|---|---|
| 1 | **2a** | 1:1.5:1.8 | DCM | rt | 60 |
| 2 | **2a** | 1:1.5:1.8 | DCE | rt | 55 |
| 3 | **2a** | 1:1.5:1.8 | $CHCl_3$ | rt | 64 |
| 4 | **2a** | 1:1.5:1.8 | MeCN | rt | 50 |
| 5 | **2a** | 1:1.5:1.8 | toluene | rt | 48 |
| 6 | **2a** | 1:1.5:1.8 | MeOH | rt | N.R |
| 7 | **2a** | 1:1.5:1.8 | THF | rt | 30 |
| 8 | **2a** | 1:1.5:1.8 | $CHCl_3$ | 10 | 68 |
| 9 | **2a** | 1:1.5:1.8 | $CHCl_3/H_2O$ (3:7) | 10 | 62 |
| 10 | **2a** | 1:1.5:1.8 | $CHCl_3/H_2O$ (1:1) | 10 | 67 |
| 11 | **2a** | 1:1.5:1.8 | $CHCl_3/H_2O$ (7:3) | 10 | 72 |
| 12 | **2a** | 1:1.5:1.8 | $CHCl_3$/pH = 6.5 buffer (7:3) | 10 | 66 |
| 13 | **2a** | 1:1.5:1.8 | $CHCl_3$/pH = 7.8 buffer (7:3) | 10 | 74 |
| 14 | **2a** | 1:1.5:1.8 | $CHCl_3$/pH = 8.0 buffer (7:3) | 10 | 81 |
| 15 | **2a** | 1:1.5:1.8 | $CHCl_3$/pH = 9.0 buffer (7:3) | 10 | 79 |
| 16 | **2b/2c/2d** | 1:1.5:1.8 | $CHCl_3$/pH = 8.0 buffer (7:3) | 10 | 55/21/trace |
| 17 | **2a** | 1:1:1 | $CHCl_3$/pH = 8.0 buffer (7:3) | 10 | 52 |
| 18 | **2a** | 1:1.2:1.8 | $CHCl_3$/pH = 8.0 buffer (7:3) | 10 | 62 |

[a]Reaction conditions: **1a** (0.2 mmol), **2a** (0.3 mmol), **3a** (0.36 mmol), and solvent (1 mL) under an argon atmosphere for 24 hours unless otherwise specified.
[b]Isolated yield.

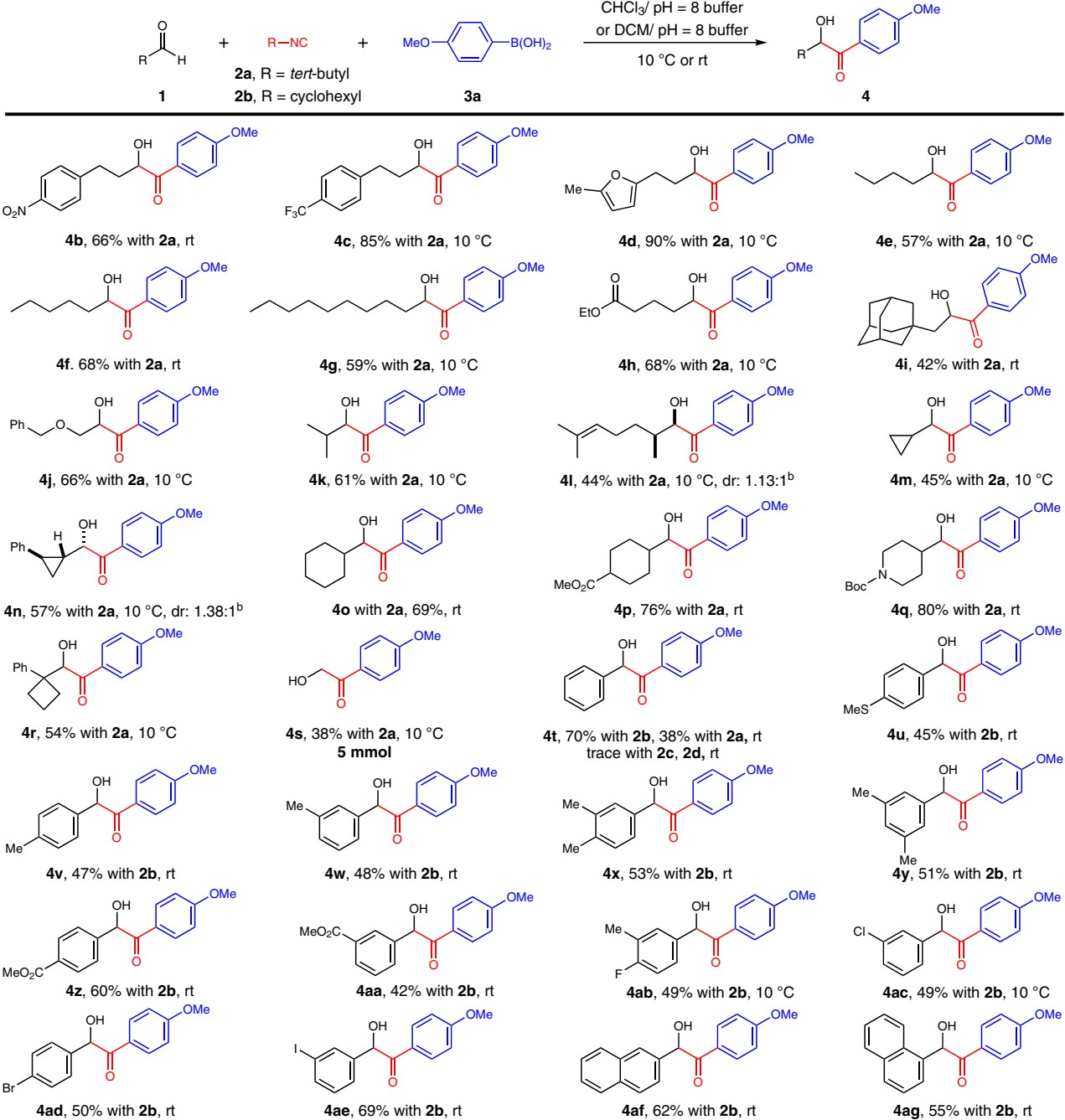

**Fig. 2 Scope of aldehydes[a].** Reaction conditions: [a]aldehyde **1** (0.2 mmol, 1 equiv), **2a** (0.3 mmol), **3a** (0.36 mmol), and CHCl₃/pH = 8 buffer (7:3, 1 mL) under an argon atmosphere for 24 hours unless otherwise specified; [b]The dr was determined by ¹H NMR analysis. [c]aldehyde **1** (0.2 mmol, 1 equiv), **2b** (0.3 mmol), **3a** (0.36 mmol), and DCM/pH = 8 buffer (7:3, 1 mL) under an argon atmosphere for 24 hours.

congeners, good reactivities were exhibited. Arylboronic acids with electronically neutral *meta-para*-dimethyl, *para*-methyl, and *para*-tertbutyl substituents produced α-hydroxyketones **5a–5c** in moderate yields. Analogs with electron-rich substituents such as acetal, alkoxy, and diphenylamino groups reacted smoothly towards products **5d–5l** in 51–85% yields. Inclusion of alkenyl or alkynyl group was noteworthy; from which products **5j** and **5k** were acquired in 80% and 77% yield. This study was auspiciously and effortlessly extendable to a series of heteroarylboronic acids containing furan (**5m**), thiophene (**5n–5p**), benzofuran (**5q**), benzothiophen (**5r**), protected or unprotected indoles (**5s, 5u**), 7-azaindole (**5t**),

dibenzothiophene (**5w**) and carbazole (**5x**) cores. Remarkably, both aryl and alkyl substituted alkenylboronic acids could rendered the corresponding α-hydroxy enones **5y** and **5z** in 82% and 63% yields, which broadly expand the scope of the products. For electron-deficient substituted boronic acids, such as the halobenzene boronic acids, only trace amounts of products could be obtained, which probably is due to their low nucleophilicity that cannot capture the nitrilium intermediates. Aliphatic boronic acids, such as phenethylboronic acid and cyclopentylboronic acid, do not react under our standard conditions, perhaps owing to the lack of π electrons which makes 1,4-alkyl shift difficult[68].

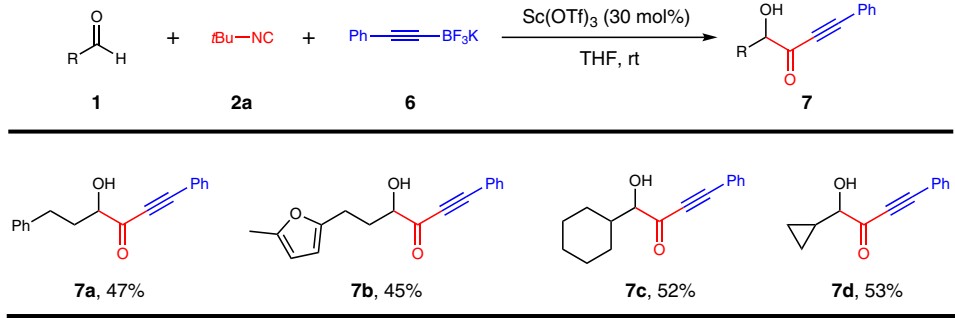

**Fig. 3 Scope of boronic acids[a].** Reaction conditions: [a]**1a** (0.2 mmol, 1 equiv), **2a** (0.3 mmol), **3** (0.36 mmol), and CHCl$_3$/pH = 8 buffer (7:3, 1 mL) under an argon atmosphere for 24 hours unless otherwise specified. [b]**1** (0.2 mmol, 1 equiv), **2a** (0.3 mmol), **3** (0.36 mmol), and DCM/pH = 8 buffer (7:3, 1 mL) under an argon atmosphere for 24 hours.

**Fig. 4 Passerini-type reaction of alkynyl trifluoroborate salt.** Reaction conditions: **1** (0.2 mmol, 1 equiv), **2a** (0.5 mmol), **6** (0.6 mmol), and THF (1.5 mL) under an argon atmosphere for 12 hours unless otherwise specified.

**Passerini-type reaction of alkynylboron compounds.** α-Hydroxy alkynylketones are important intermediates for the synthesis of natural products and drug molecules[82,83]. However, the synthesis of such α-hydroxyketones has faced significant challenges and usually multiple steps are required[82,83]. We sought to explore the Passerini-type reaction on alkynylboron compounds, if successful, a straightforward and efficient method could be disclosed for the synthesis of α-hydroxy alkynylketones, which further demonstrates the strengths and capability of our protocol (Fig. 4). Alkynyl trifluoroborate salt was employed as the source of alkyne in our transformation owing to the instability of alkynylboronic acid. To our delight, the Passerini-type reaction of alkynyl

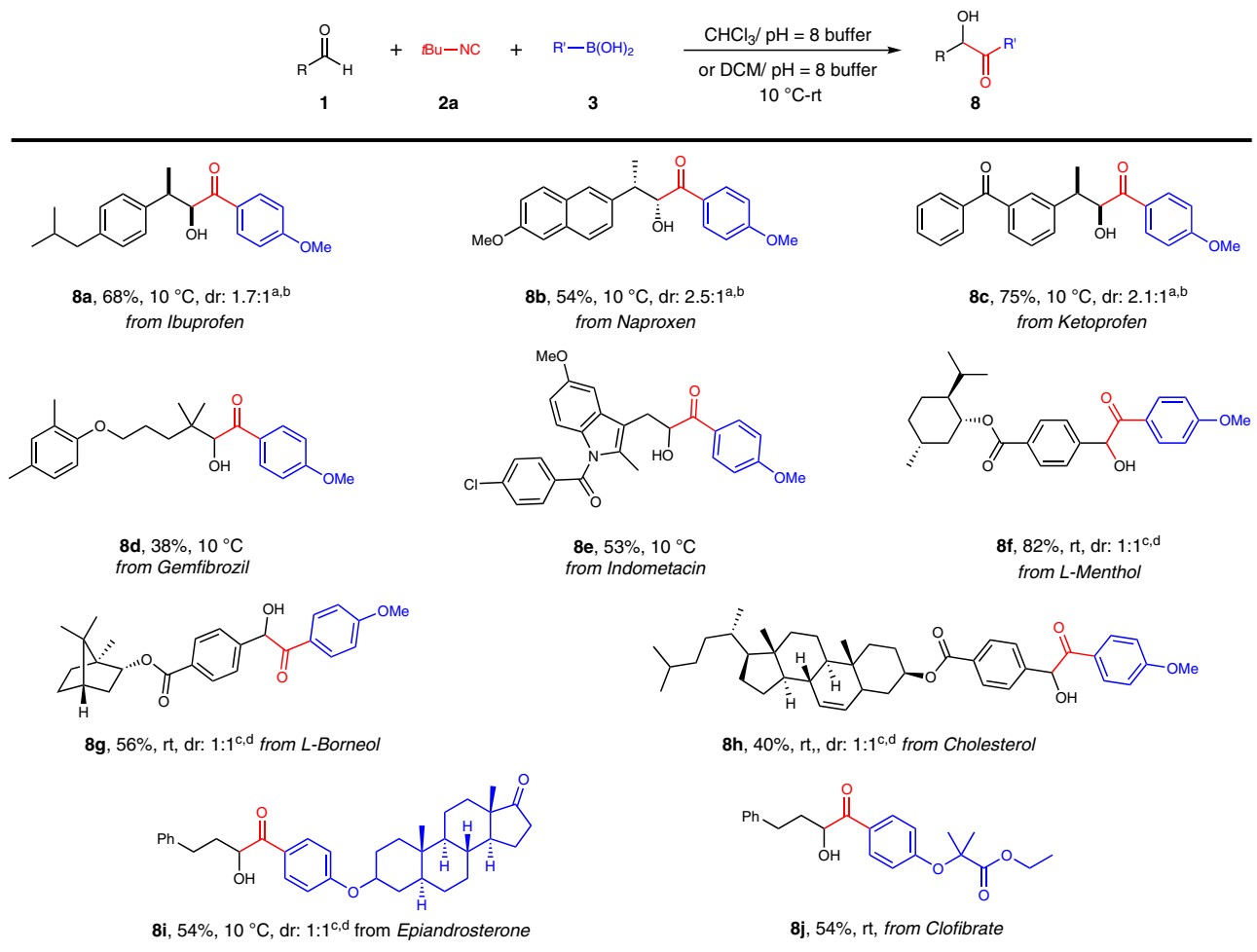

**Fig. 5 Late-stage modifications of bioactive or drug molecules.** Reaction conditions: [a]**1a** (0.2 mmol, 1 equiv), **2a** (0.3 mmol), **3** (0.36 mmol), and CHCl₃/pH = 8 buffer (7:3, 1 mL) under an argon atmosphere for 24 hours unless otherwise specified. [b]The dr was determined by ¹H NMR analysis. [c]**1** (0.2 mmol, 1 equiv), **2a** (0.3 mmol), **3** (0.36 mmol) and DCM/pH = 8 buffer (7:3, 1 mL) under an argon atmosphere for 24 hours. [d]The dr was determined by HPLC analysis.

trifluoroborate salt could proceed smoothly under the action of Lewis acid (Sc(OTf)₃), and the target products (**7a–7d**) could be obtained in a moderate yield. This reaction could not occur without Lewis acid (Sc(OTf)₃), probably because a four-coordinated boron "ate" nitrilium intermediate could not be generated from potassium phenyltrifluoroborate, aldehyde, and isocyanide. Lewis acid may promote the conversion of potassium phenyltrifluoroborate (**6**) into phenyldifluoroborane, which could form a four-coordinated boron intermediate[84].

**Late-stage modifications of complex molecules.** The excellent functional group compatibility prompted our endeavors to extrapolate this synthesis scheme to late-stage modification of bioactive or therapeutic agents (Fig. 5). A series of bioactive or drug molecules (Ibuprofen, Naproxen, Ketoprofen, Gemfibrozil, Indometacin, L-Menthol and L-Borneol, and Cholesterol) were derivatized into corresponding aldehydes which, upon treatment with 4-methoxyphenyl boronic acid under established Passerini-type coupling conditions, were smoothly incorporated in eventual α-hydroxyketone derivatives **8a–8h**. Futhermore, conversions of arylboronic acids that were derived from drug molecules such as Epiandrosterone and Clofibrate had brought forth drug analogs **8i** and **8j** in moderate yields. It was thus envisioned that this method would simplify access to discover other bioactive molecules. The previous MCRs involving isocyanide exhibit poor

stereoselectivity. This Passerini-type reaction of boronic acids showed similar results in terms of stereochemical control. In most cases (**4l**, **4n**, **8a–8c**, and **8f–8i**), the dr values remained between 1:1 and 2.5:1 (see the Supplementary Information for details).

**Gram-scale synthesis and synthetic applications.** The practical constraint of this Passerini-type MCR with boronic acids was next evaluated through translation to gram-scale synthesis. As shown in Fig. 6a, reaction efficiencies were preserved on 2 gram-scale (10 mmol, 50 times), thus implying the application potential for industrial production of the α-hydroxyketones.

The readiness of α-hydroxyketone products for chemical manipulations was pronounced in production of 1,2-diol (**9**), 1,2-dione (**10**), quinoxaline (**11**), cyclic sulfamate imine (**12**), and poly-substituted oxazole (**13**) (Fig. 6b). The innate step economy of MCRs has also presented an abbreviated route towards (±) Harmandianone, a phenylpropanoid derivative isolated from Clausena Harmandiana fruits[85], from simple building blocks (Fig. 6c). Of further significance, products could be precursors for entities that constitute the structural core of bioactive compounds such as α-acyloxy lactone and α,β-unsaturated lactone. The former (**19**) was fabricated upon lactonization of α-hydroxyketone **18** formed from methyl 4-oxobutanoate (**17**) (Fig. 6d). A two-step olefination and ring-closing olefin metathesis of **4a** afforded α,β-unsaturated lactone product (±) **22** (Fig. 6e). The

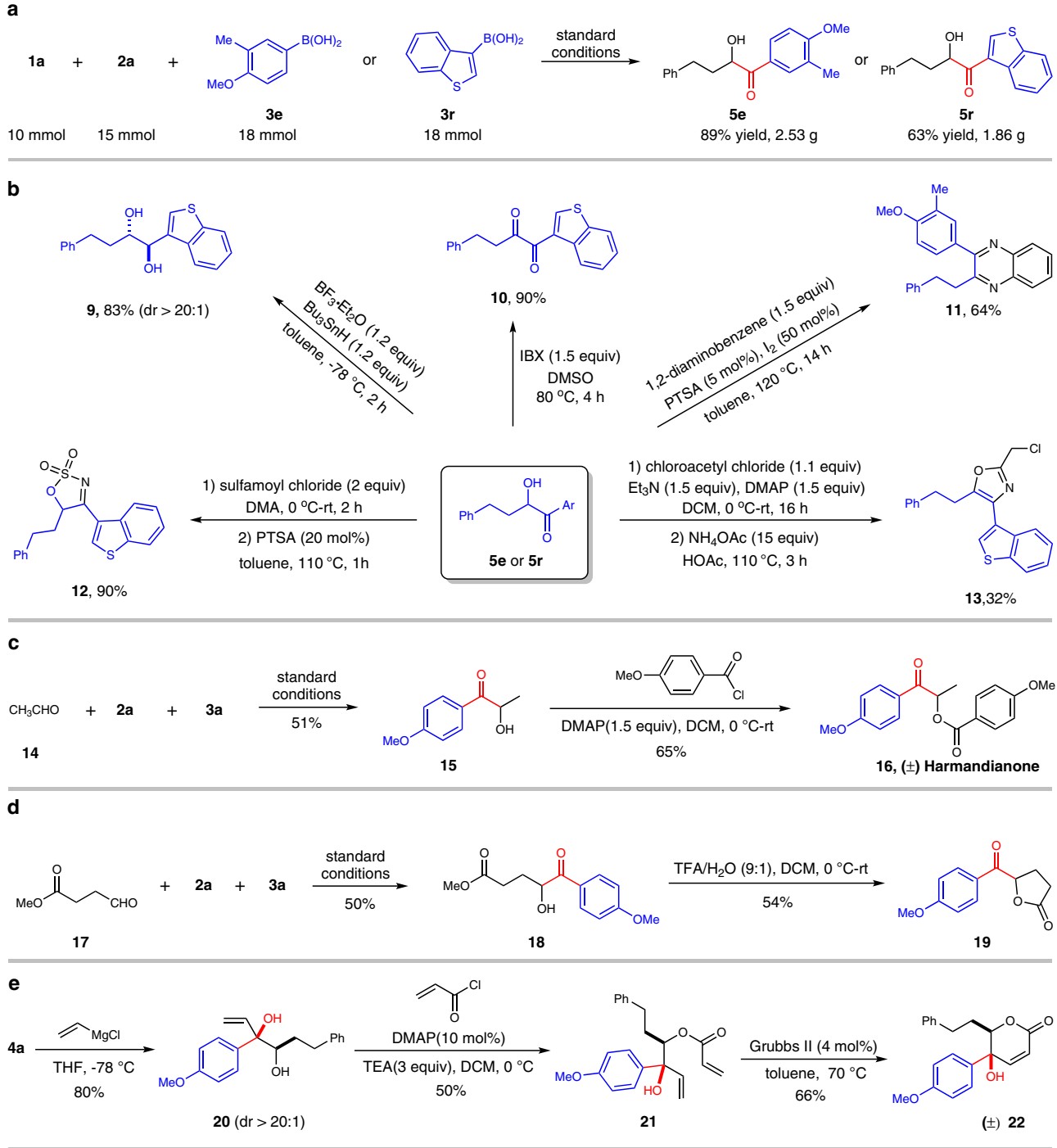

**Fig. 6 Gram-scale synthesis and synthetic applications. a** Gram-scale synthesis. **b** Transformations of α-hydroxyketones. **c** Synthesis of Harmandianone. **d** Synthesis of α-acyloxy lactone. **e** Synthesis of α, β-unsaturated lactone.

high diastereoselectivity of compound **20** may originate from the chelation of hydroxyl, carbonyl oxygen, and magnesium[86].

In conclusion, we have realized the application of boronic acid as carbon nucleophiles in the manifold of Passerini reaction. Accordingly, this protocol provided simplified modular access of α-hydroxyketones from aldehydes, isocyanide, and boronic acids. The functional group tolerance of this chemistry has supported late-stage diversifications of bioactive products and pharmaceuticals through this three-component coupling reaction. The wealth of follow-up chemical conversions that could be

performed on procured α-hydroxyketones has additionally illustrated the utility of this method.

## Methods

**General procedure A for the synthesis of α-hydroxyketones from alkylalde-hydes**. In air, a 10 mL schlenk tube was charged with arylboronic acids (0.36 mmol, 1.8 equiv). The tube was evacuated and filled with argon for three cycles. Then, chloroform (0.7 mL), pH = 8 buffer (0.3 mL), alkylaldehydes (0.20 mmol, 1 equiv), tertbutyl isocyanide (34 μl, 0.30 mmol, 1.5 equiv) were added under argon. The reaction was allowed to stir at corresponding temperature for 24 hours. Upon completion, proper amount of silica gel was added to the reaction mixture. After

removal of the solvent, the crude reaction mixture was purified on silica gel (petroleum ether and ethyl acetate) to afford the desired products.

**General procedure B for the synthesis of α-hydroxyketones from arylaldehydes.** In air, a 10 mL schlenk tube was charged with arylboronic acids (0.36 mmol, 1.8 equiv). The tube was evacuated and filled with argon for three cycles. Then, dichloromethane (0.7 mL), pH = 8 buffer (0.3 mL), arylaldehydes (0.20 mmol, 1 equiv), cyclohexyl isocyanide (37 μl, 0.30 mmol, 1.5 equiv) were added under argon. The reaction was allowed to stir at room temperature for 24 hours. Upon completion, proper amount of silica gel was added to the reaction mixture. After removal of the solvent, the crude reaction mixture was purified on silica gel (petroleum ether and ethyl acetate) to afford the desired products.

**General procedure C for the synthesis of α-hydroxyketones from alkynyl trifluoroborate salt.** In air, a 10 mL schlenk tube was charged with alkynyl trifluoroborate salt (0.60 mmol, 3 equiv) and Sc(OTf)$_3$ (30.0 mg, 0.06 mmol, 0.3 equiv). The tube was evacuated and filled with argon for three cycles. Then, THF (1.5 mL), aldehydes (0.20 mmol, 1 equiv), and tertbutyl isocyanide (57 μl, 0.50 mmol, 2.5 equiv) were added under argon. The reaction was allowed to stir at room temperature for 12 hours. Upon completion, proper amount of silica gel was added to the reaction mixture. After removal of the solvent, the crude reaction mixture was purified on silica gel (petroleum ether and ethyl acetate) to afford the desired products.

## Data availability

The data supporting the finding of this study are available within the paper and its Supplementary Information.

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

## Acknowledgements

Financial supports from the National Natural Science Foundation of China (21772046, 2193103) are gratefully acknowledged.

## Author contributions

Q.S. conceived the project. K.Y., F.Z., T.F., C.L., and W.L. performed experiments and prepared the Supplementary Information. Q.S. and K.Y. prepared the manuscript. All authors discussed the results and commented on the manuscript.

## Competing interests

The authors declare no competing interests.
