## [Peer Review File · Nature Communications]

REVIEWER COMMENTS

Reviewer #1 (Remarks to the Author):

Song and co-workers presented a new contribution to the chemistry of isocyanides where a C-nucleophile is able to trap intramolecularly an isonitrilium ion derived from the interaction between the isocyanide and an activated carbonyl compound. The results are interesting and deserve to be published but I think that a more specialized organic chemistry journal is more suited. However, before submitting the paper elsewhere the authors should properly revise the manuscript because some parts are confused and the SI need to be carefully reviewed and improved.

Some suggestions:

- The main drawback of this new method is the poor atom economy, as the substituent of the isocyanide is lost during work-up. The authors say that "It was further established that on replacement of tert-butyl isocyanide (2a) with cyclohexyl isocyanide (2b), benzyl isocyanide (2c) or ethyl 2-isocyanoacetate (2d), formation efficiency of α -hydroxyketone product 4a was diminished." This sentence makes sense if a comparison between the yields using different isocyanides are reported. Moreover, in Table 2 CyNC has been used: this means that its behavior is not too bad. I suggest to try methyl isocyanide because in this case only 1 carbon atom is lost. If the yields are not good this can explain why t-BuNC is preferred, which should be outlined in the paper.
- Table 2 is confused: for every new synthesized molecule the used isocyanide must be evidenced. Moreover, it is necessary to explain in the text that, using aromatic aldehydes CyNC was used instead of tBuNC and at least a couple of experiments with tBuNC should be done with aromatic aldehydes for comparing the efficiency of the two isocyanides.
- pH has been not correctly written either in the main text or in the SI.
- The authors report that aliphatic boronic acids are poor coupling partners. However, which ones were used and which was the yield? Or you mean that the reaction does not work or a decomposition occurred? This must be explained.
- Tables 2, 3, 4, 5 (the last one was numbered 4 by the authors, which is clearly an error) are more suited as Schemes instead of tables.
- The sentence just above Figure 3 must be checked.
- Figure 4, part A: this part of the figure is confused, because not all transformations are obvious. Only in this case the reagents are reported in the captions and many of them are missing. I suggest different schemes for every transformation, some additional comments in the text and all reagents over the arrows for transformation A. Remember: BF₃·Et₂O, not BF₃-Et₂O.
- Every time you prepared compounds with 2 stereogenic centers it is important to report the d.r.: this has been done for compounds 8a,c, but not for 8b,d,g,h,i,j. I expect a poor stereocontrol for compounds 8g-j, but in compounds 8b,d as well it is unlikely that just one diastereoisomer is obtained, as it seems from the structure reported in Figure 4. Please, clarify. Comments on the stereoselectivity and related data, supported by analytic data on the method used for d.r. determination must be provided. Moreover, the authors should specify if the two diastereoisomers have been separated (or if they cannot be separated by chromatography) and if only one of them or the mixture was characterized. Of course, for separated diastereoisomers the specific optical rotation must be provided.
- Compound 20: it seems that only one diastereoisomer is obtained: please specify the d.r. and describe how was the relative stereochemistry determined.

I have found many criticisms in the SI, which has to be improved:

- General procedure A: why is only tertbutyl isocyanide written? Both tBuNC and CyNC have been used. I'm very surprised by the procedure used for work-up which is unusual and not correct in my opinion: if a reaction is performed in a two phase system, and the buffer is 30% of the whole mixture, the aqueous phase has to be removed by extraction before chromatography. The reported work-up is justified only if the overall performance is better but this should be explained.
- I suggest to change phenylpropyl aldehyde into 3-phenylpropanal (throughout the SI).
- The names of some aldehydes are incorrect. See: 4g,p and 5c.
- Check the names of boronic acids. See: 5c-t, 5v-x, 8k.
- Compound 4u: report the number after the name.
- In the general procedure it is correct to simply mention the eluent, but for every compound the information has to be given, along with the TLC analysis and R_f.

- I have noticed that no IR spectra are reported.
- For all solid compounds the melting point has to be measured. See: 4ac, 5c, 5u, 5v, 5x, 8b, 9, 12.
- Transformation of 5r into 13: chloroacetyl chloride is missing over the arrow.
- Check also for some typos.

The manuscript describes an easy and efficient approach to α -hydroxyketones via a novel Passerini-type reaction using boronic acids. The methodology was optimized using model compounds (phenylpropyl aldehyde, *tert*-butyl isocyanide and 4-methoxyphenylboronic acid) by modifications in the temperature and solvent system. The ideal conditions were found to be a 7:3 mixture of CHCl_3 /pH 8.0 buffer at 10 °C. Next, under these conditions, the reaction scope was studied by varying the aldehydes and boronic acids, and a great variety of products could be obtained in moderate to very good yields. Different functional groups were well tolerated in the reactions and α -hydroxyketones bearing aromatic as well as aliphatic groups were easily obtained. Noteworthy is the application of this methodology to the synthesis of α -hydroxy alkynylketones, which are challenging to prepare by other methods. The reactions were carried out on a very small scale (0.2 mmol) but the process was scaled-up for 2 examples and worked well on 2-gram scale. Derivatives of bioactive and drug molecules could also be efficiently obtained through this method. Furthermore, the authors describe some synthetic applications of this method, including the 2-step synthesis of racemic Harmandianone, a new phenylpropanoid derivative. In conclusion, I consider this manuscript of great relevance to the people on the field once α -hydroxyketones are useful building blocks in organic synthesis and the developed procedure allows easy access to these molecules. Thus, I recommend its publication in Nature Communications. Nevertheless, there are some issues that should be addressed by the authors and corrections to be made as pointed out below:

1. Page 3, 1st line: “thus mediating” instead of “thus mediates”; 2nd paragraph, 3rd line from the bottom: “thus promoting” instead of “thus promotes”.
2. Page 3, 2nd paragraph: do the authors have any explanation for the complete failure of MeOH as solvent in this reaction?
3. Page 3, 2nd paragraph: the authors stated that replacement of *tert*-butyl isocyanide with other isocyanides lowered the efficiency of the reactions. These results are not shown in the table and should be added. Nevertheless, table 2 shows many examples in which cyclohexyl isocyanide **2b** was used with success. The authors should check that out.
4. Page 4, table 1: regarding the relative ratio between the reagents (0.2 mmol of **1a**, 0.3 mmol of **2a** and 0.36 mmol of **3a**), the authors should comment whether they have tried other possibilities, including an equimolar amount of these reagents (that should be ideal).
5. Page 4, table 1, entries 12-15: “pH” instead of “PH”. Line 11: “paraformaldehyde” instead of “formaldehyde”; line 13: “diverse” instead of “divers”.
6. Page 5, table 2: specify the stereochemistry of compound **4n** as the *trans* or *cis* isomer with respect to the substituents on the cyclopropane ring.
7. Page 6, table 3: were electron-poor arylboronic acids investigated? In this table, rotate the molecules 180° such that they keep the standard used in other tables with the blue portion to the right.
8. Page 7, table 4: what happens in the absence of $\text{Sc}(\text{OTf})_3$ in those reactions? Why is it necessary? The authors should comment on that.
9. Page 8: table 4 should be table 5. In this table, the dr is only shown for compounds **8a** and **8c**. Where appropriable, the dr values for the other compounds should be given. Interestingly, looking at the spectra of these compounds, it seems that only one isomer was obtained. If this is the case, the authors should comment on that and try to explain the high selectivity. Also, the carbonyl group attached to the Ar group in compound **8g** should be in red color. Compound **8k** should be rotated (see comment number 7).
10. Page 8, line 3: “thus implying” instead of “thus implied”.

11. Figures 3 and 4, on pages 8 and 9, should be schemes, not figures.
12. Page 9, Figure 4: many things are missing in the synthetic schemes. What is the relative stereochemistry of compound **9**? It is represented as *syn* in the supporting info. *o*-aminoaniline, sulfamoyl chloride and chloroacetyl chloride are missing as reagents in steps c, d and e, respectively. The dr for compound **20** should be given.
13. In the reference section, many things should be corrected, including: scientific names on references 1, 2, 5 and 84 should be written accordingly (italics and first name in capital letters); many proper names throughout the section should begin with capital letters such as Ugi, Passerini, Heyns, Pudovik, Bronsted, Lewis, Organic Synthesis, Schiff, Smiles, Suzuki, Mannich, Petasis, etc. Ga and Yb as well. Ref. 13: BF₃.OEt₂; Ref. 26: RuO₄; Ref. 55: remove 1,2 after "mechanism"; Ref. 73: C(sp³)-C(sp²) instead of c(sp³)-c(sp²).
14. Supporting information:
 - 14.1. General information: insert information about the source of commercial reagents.
 - 14.2. General procedures A and B: line 1, "0.36 mmol" instead of "0.38 mmol"; "0.20 mmol" and "0.30 mmol" instead of "0.2 mmol" and "0.3 mmol" (the same for all compounds **4** in the following pages); "were added" instead of "was added" (same for general procedure C); the ratio of solvents used in the chromatographic purifications should be added (same for general procedure C).
 - 14.3. Page 2, from compound **4b** on: multiplets are represented by a range within which the dashes should be uniform (use either – or -). This lack of uniformity can be seen in the whole experimental section.
 - 14.4. Page 3: compound **4g** was prepared from decanal and not from phenylpropyl aldehyde.
 - 14.5. Page 4, compound **4i**: "(3*R*,5*R*,7*R*)" instead of "(3*r*,5*r*,7*r*)".
 - 14.6. Page 6: state the stereochemistry of **4n**.
 - 14.7. Pages 11 and 12: compounds **5a** and **5b** are switched (see table 3). Rotate all compounds **5** 180° such that they keep the standard used in other parts of the manuscript with the blue portion to the right (same in the spectra).
 - 14.8. Pages 12-18, compounds **5c-5x**: put the correct boronic acid for each case and correct the amounts used and the yields obtained according to table 3. (4-Methoxyphenyl)boronic acid was wrongly put in all cases.
 - 14.9. Page 13, **5e**: wrong value for HRMS found.
 - 14.10. Page 21, **8b**: melting point and optical rotation are missing.
 - 14.11. Page 22: rotate **8f** by 180°.
 - 14.12. Page 24, **8i**: melting point is missing; rotate **8k** by 180°.
 - 14.13. Page 25, line 3: "were added" instead of "was added"; lines 4-5: "Upon completion, the reaction was diluted with CH₂Cl₂ (volume?) and the organic phases were washed with brine (volume?) and dried over Na₂SO₄." The ratio of eluents should be given for the column chromatography (same for 2nd paragraph).
 - 14.14. Page 25, 2nd paragraph: "cooled saturated NaHCO₃" (include the temperature and volume); the melting point of the product should be added; compound **9**: state in its name that the compound is racemic.
 - 14.15. Include the volume of extraction solvents and aqueous solutions used to wash the organic phases for all compounds from now on. Additionally, the ratio of eluents should be given for the column chromatography in each case. For solid compounds such as **11**, **12**, **15**, **19**, **21** and **22**, the melting points are missing (compare with literature data when appropriate).

- 14.16. Page 27: place the chloroacetyl chloride in the first step of the scheme.
- 14.17. Page 29, line 5: "were added" instead of "was added"; "The mixture was cooled to (include the temperature)"; line 6: "overnight" instead of "for overnight"; line 7: "Upon completion, the reaction was diluted" instead of "Upon completion of the reaction, diluted".
- 14.18. Page 31, NMR spectra: include captions for all spectra.

Carlos Kleber Z. Andrade

Reviewer #3 (Remarks to the Author):

This manuscript reports a novel synthesis of α -hydroxyketones through a three components reaction which can be seen as a combination of Paserini and Petasis reactions. Although both classes of reactions have been extensively studied for many years, to the best of my knowledge, the multicomponent reactions of carbonyls, isocyanides and boronic acids that is described in this paper have not been previously reported. The methodology reported here is fairly general, opening the door to the synthesis of a very wide variety of the synthetically valuable α -hydroxy ketones, in a straightforward manner, from readily available starting materials and in an operationally simple and metal-free procedure.

The scope of the reaction has been studied in detail, as illustrated by the large number of examples included. Additionally, the methodology has been applied to a number of challenging substrates, such as modified drugs and natural products.

I consider that this methodology will be very useful in the context of organic synthesis, and for this reason I think it reaches the high standards to be published in Nature Communications.

The supplementary information is complete and the compounds are well characterized.

Some minor points:

- The reaction with aryl boronic acids is restricted to electron-rich or electron-neutral systems. Nothing is said about electron-poor derivatives, such as the halobenzene boronic acids. Some commentary should be included.
- It is curious that the reactions with the aldehyde derived from ibuprofene and ketoprofene give rise a to mixture of two diastereoisomes, but the reaction with the analogous naproxene derivative (which is structurally very similar) provides only one isomer. Does the reaction crude present only one stereoisomer in this case? And how was the stereochemistry proposed determined?
- In figure 4, A, there are some reagent missing for the synthesis of 11 and 12.

REVIEWER COMMENTS

Reviewer #1 (Remarks to the Author):

Song and co-workers presented a new contribution to the chemistry of isocyanides where a C-nucleophile is able to trap intramolecularly an isonitrilium ion derived from the interaction between the isocyanide and an activated carbonyl compound. The results are interesting and deserve to be published but I think that a more specialized organic chemistry journal is more suited.

Response: Thank you for your comments on our work.

However, before submitting the paper elsewhere the authors should properly revise the manuscript because some parts are confused and the SI need to be carefully reviewed and improved.

Some suggestions:

- The main drawback of this new method is the poor atom economy, as the substituent of the isocyanide is lost during work-up. The authors say that "It was further established that on replacement of *tert*-butyl isocyanide (**2a**) with cyclohexyl isocyanide (**2b**), benzyl isocyanide (**2c**) or ethyl 2-isocyanoacetate (**2d**), formation efficiency of α -hydroxyketone product **4a** was diminished." This sentence makes sense if a comparison between the yields using different isocyanides are reported. Moreover, in Table 2 CyNC has been used: this means that its behavior is not too bad. I suggest to try methyl isocyanide because in this case only 1 carbon atom is lost. If the yields are not good this can explain why *t*-BuNC is preferred, which should be outlined in the paper.

Response: Thank you for pointing it out. The results of other isocyanides (**2b**, **2c**, **2d**) have been added to Table 1, and their yields are all lower than *tert*-butyl isocyanide (**2a**). However, cyclohexyl isocyanide (**2b**) is more suitable for reactions involving aromatic aldehydes (**4t-4ag**, Scheme 1; previous Table 2).

Isocyanides (**2a**, **2b**, **2c**, **2d**) are commercially available. Methyl isocyanide has not

been commercialized yet, its boiling point is only 59.6 °C, and it emits a very uncomfortable smell, and it is carcinogenic, therefore it is not conducive to employ methyl isocyanide in this reaction.

- Table 2 is confused: for every new synthesized molecule the used isocyanide must be evidenced. Moreover, it is necessary to explain in the text that, using aromatic aldehydes CyNC was used instead of tBuNC and at least a couple of experiments with tBuNC should be done with aromatic aldehydes for comparing the efficiency of the two isocyanides.

Response: Thank you for pointing it out. The isocyanide used for each synthesized new compound has been labeled in Scheme 1 (previous Table 2).

Benzaldehyde can only get a 38% yield product **4t** with *tert*-butyl isocyanide (**2a**). We tried other isocyanates (**2b**, **2c**, **2d**) and found that cyclohexyl isocyanide (**2b**) can get a higher yield (70%) for the same product. Therefore, we used cyclohexyl isocyanide (**2b**) to examine the scope of aromatic aldehydes.

Please see our revised manuscript.

- pH has been not correctly written either in the main text or in the SI.

Response: Thank you for pointing it out, all 'PH's have been changed to 'pH's, please see our revised manuscript.

- The authors report that aliphatic boronic acids are poor coupling partners. However, which ones were used and which was the yield? Or you mean that the reaction does not work or a decomposition occurred? This must be explained.

Response: Thank you for pointing it out. These results have been commented in the main text, please see our revised manuscript. Aliphatic boronic acids, such as phenethylboronic acid does not react with cyclopentyl aldehyde under the standard condition, it might be due to the lack of π electrons which makes 1,4-shift difficult for alkyl boronic acids.

- Tables 2, 3, 4, 5 (the last one was numbered 4 by the authors, which is clearly an error) are more suited as Schemes instead of tables.

Response: Thank you for pointing it out, Tables 2, 3, 4, 5 have been changed to Scheme 1, 2, 3, 4, please see our revised manuscript.

- The sentence just above Figure 3 must be checked.

Response: Thank you for pointing it out, the sentence just above Scheme 5 (previous Figure 3) has been revised, please see our revised manuscript.

- Figure 4, part A: this part of the figure is confused, because not all transformations are obvious. Only in this case the reagents are reported in the captions and many of them are missing. I suggest different schemes for every transformation, some additional comments in the text and all reagents over the arrows for transformation A. Remember: $\text{BF}_3 \cdot \text{Et}_2\text{O}$, not $\text{BF}_3\text{-Et}_2\text{O}$.

Response: Thank you for pointing it out. The Scheme 6A (previous Figure 4A) has been modified and all reagents have been added to the arrows, please see our revised manuscript. And $\text{BF}_3\text{-Et}_2\text{O}$ has been corrected to $\text{BF}_3 \cdot \text{Et}_2\text{O}$.

- Every time you prepared compounds with 2 stereogenic centers it is important to report the d.r.: this has been done for compounds 8a,c, but not for 8b,d,g,h,i,j. I expect a poor stereocontrol for compounds 8g-j, but in compounds 8b,d as well it is unlikely that just one diastereoisomer is obtained, as it seems from the structure reported in Figure 4. Please, clarify. Comments on the stereoselectivity and related data, supported by analytic data on the method used for d.r. determination must be provided. Moreover, the authors should specify if the two diastereoisomers have been separated (or if they cannot be separated by chromatography) and if only one of them or the mixture was characterized. Of course, for separated diastereoisomers the specific optical rotation must be provided.

Response: Thank you for pointing it out.

dr values of **8a-8c** are determined by the ^1H NMR spectrum

dr values of **8f-8i** are determined by the chiral HPLC.

The dr of **8b** (dr: 2.5:1), **8f** (dr: 1:1), **8g** (dr: 1:1), **8h** (dr: 1:1), **8i** (dr: 1:1) have been added.

Compounds **8b** is single diastereoisomer isolated, and the optical rotations have been added.

The previous compound **8d** has been deleted because we have no suitable method to determine the diastereoselectivity.

Please see our revised manuscript and SI.

- Compound 20: it seems that only one diastereoisomer is obtained: please specify the d.r. and describe how was the relative stereochemistry determined.

Response: Thank you for pointing it out. dr of compound 20 has been added. The high diastereoselectivity of compound **20** may originate from the chelation of hydroxyl, carbonyl oxygen and magnesium. Please see our revised manuscript.

I have found many criticisms in the SI, which has to be improved:

- General procedure A: why is only tertbutyl isocyanide written? Both tBuNC and CyNC have been used. I'm very surprised by the procedure used for work-up which is unusual and not correct in my opinion: if a reaction is performed in a two phase system, and the buffer is 30% of the whole mixture, the aqueous phase has to be removed by extraction before chromatography. The reported work-up is justified only if the overall performance is better but this should be explained.

Response: Thank you for pointing it out.

General procedure A is mainly used for the reaction of alkylaldehydes, tBuNC and arylboronic acid, while General procedure B is used for the reaction of arylaldehydes, CyNC and arylboronic acids.

We performed extraction and separation for gram-scale synthesis (page 29 in SI), and no extraction is required before column chromatography for the standard reactions (0.2 mmol).

Please see our revised SI.

- I suggest to change phenylpropyl aldehyde into 3-phenylpropanal (throughout the SI).

Response: Thank you for pointing it out. Phenylpropyl aldehyde has been changed to 3-phenylpropanal, please see our revised SI.

- The names of some aldehydes are incorrect. See: 4g,p and 5c.

Response: Thank you for pointing it out. Above errors have been corrected, please see our revised SI.

- Check the names of boronic acids. See: 5c-t, 5v-x, 8k.

Response: Thank you for pointing it out. Above errors have been corrected, please see our revised SI.

- Compound 4u: report the number after the name.

Response: Thank you for pointing it out. The “4u” has been added, please see our revised SI.

- In the general procedure it is correct to simply mention the eluent, but for every compound the information has to be given, along with the TLC analysis and R_f.

Response: Thank you for pointing it out. TLC analysis and R_f of all compounds have been added, please see our revised SI.

- I have noticed that no IR spectra are reported.

Response: Thank you for pointing it out. The IR characteristic peaks of carbonyl and hydroxyl have been added, please see our revised SI.

- For all solid compounds the melting point has to be measured. See: 4ac, 5c, 5u, 5v, 5x, 8b, 9, 12.

Response: Thank you for pointing it out. The melting points of all solid compounds

have been added, please see our revised SI.

- Transformation of 5r into 13: chloroacetyl chloride is missing over the arrow.

Response: Thank you for pointing it out. Chloroacetyl chloride has been added, please see our revised SI.

- Check also for some typos.

Response: Thank you for pointing it out. Some typos have been checked, please see our revised SI.

Reviewer #2 (Remarks to the Author):

The manuscript describes an easy and efficient approach to α -hydroxyketones via a novel Passerini-type reaction using boronic acids. The methodology was optimized using model compounds (phenylpropyl aldehyde, tert-butyl isocyanide and 4-methoxyphenylboronic acid) by modifications in the temperature and solvent system. The ideal conditions were found to be a 7:3 mixture of $\text{CHCl}_3/\text{pH } 8.0$ buffer at 10°C . Next, under these conditions, the reaction scope was studied by varying the aldehydes and boronic acids, and a great variety of products could be obtained in moderate to very good yields. Different functional groups were well tolerated in the reactions and α -hydroxyketones bearing aromatic as well as aliphatic groups were easily obtained. Noteworthy is the application of this methodology to the synthesis of α -hydroxy alkynylketones, which are challenging to prepare by other methods. The reactions were carried out on a very small scale (0.2 mmol) but the process was scaled-up for 2 examples and worked well on 2-gram scale. Derivatives of bioactive and drug molecules could also be efficiently obtained through this method. Furthermore, the authors describe some synthetic applications of this method, including the 2-step synthesis of racemic Harmandianone, a new phenylpropanoid derivative. In conclusion, I consider this manuscript of great relevance to the people on the field once α -hydroxyketones are useful building blocks in organic synthesis

and the developed procedure allows easy access to these molecules. Thus, I recommend its publication in Nature Communications.

Response: Thank you for your favorable comments on our work, we really appreciate it.

Nevertheless, there are some issues that should be addressed by the authors and corrections to be made as pointed out below:

1. Page 3, 1st line: “thus mediating” instead of “thus mediates”; 2nd paragraph, 3rd line from the bottom: “thus promoting” instead of “thus promotes”.

Response: Thank you for pointing it out, these errors have been corrected, please see our revised manuscript.

2. Page 3, 2nd paragraph: do the authors have any explanation for the complete failure of MeOH as solvent in this reaction?

Response: Thank you for pointing it out. The reaction of arylboronic acid ($\text{ArB}(\text{OH})_2$) with methanol may form arylborate ($\text{ArB}(\text{OMe})_2$), which is not conducive to the formation of boron “ate” complex and the subsequent 1,4- aryl shift.

3. Page 3, 2nd paragraph: the authors stated that replacement of *tert*-butyl isocyanide with other isocyanides lowered the efficiency of the reactions. These results are not shown in the table and should be added. Nevertheless, table 2 shows many examples in which cyclohexyl isocyanide **2b** was used with success. The authors should check that out.

Response: Thank you for pointing it out. The results of other isocyanides (**2b**, **2c**, **2d**) have been shown in Table 1.

Benzaldehyde can only get a 38% yield product **4t** with *tert*-butyl isocyanide (**2a**). We tried other isocyanates (**2b**, **2c**, **2d**) and found that cyclohexyl isocyanide (**2b**) can get a higher yield (70%) product. Therefore, we used cyclohexyl isocyanide (**2b**) to examine the scope of aromatic aldehydes.

Please see our revised manuscript.

4. Page 4, table 1: regarding the relative ratio between the reagents (0.2 mmol of **1a**, 0.3 mmol of **2a** and 0.36 mmol of **3a**), the authors should comment whether they have tried other possibilities, including an equimolar amount of these reagents (that should be ideal).

Response: Thank you for pointing it out. Other ratios of reagents have been added to Table 1. Please see our revised manuscript.

5. Page 4, table 1, entries 12-15: “pH” instead of “PH”. Line 11: “paraformaldehyde” instead of “formaldehyde”; line 13: “diverse” instead of “divers”.

Response: Thank you for pointing it out. Above errors have been corrected, please see our revised manuscript.

6. Page 5, table 2: specify the stereochemistry of compound **4n** as the trans or cis isomer with respect to the substituents on the cyclopropane ring.

Response: Thank you for pointing it out. The substituents on the cyclopropane ring of compound **4n** is trans. please see our revised manuscript.

7. Page 6, table 3: were electron-poor arylboronic acids investigated? In this table, rotate the molecules 180° such that they keep the standard used in other tables with the blue portion to the right.

Response: Thank you for pointing it out. We have tried electron-deficient substituted boronic acids, such as the halophenyl boronic acids, only trace amounts of products can be obtained. The molecular structure has been rotated, please see our revised manuscript.

8. Page 7, table 4: what happens in the absence of Sc(OTf)₃ in those reactions? Why is it necessary? The authors should comment on that.

Response: Thank you for pointing it out. This reaction could not occur without Lewis acid (Sc(OTf)₃), probably because a four-coordinated boron “ate” nitrilium

intermediate could not be generated from potassium phenyltrifluoroborate, aldehyde and isocyanide. Lewis acid may promote the conversion of potassium phenyltrifluoroborate (**6**) into phenyldifluoroborane which could form a four-coordinated boron intermediate. Please see our revised manuscript.

9. Page 8: table 4 should be table 5. In this table, the dr is only shown for compounds **8a** and **8c**. Where appropriate, the dr values for the other compounds should be given. Interestingly, looking at the spectra of these compounds, it seems that only one isomer was obtained. If this is the case, the authors should comment on that and try to explain the high selectivity. Also, the carbonyl group attached to the Ar group in compound **8g** should be in red color. Compound **8k** should be rotated (see comment number 7).

Response: Thank you for pointing it out, this is our negligence.

The dr of naproxene derivative is 2.5:1.

Compounds **8b** and **8d** are single diastereoisomers isolated.

The dr values of **8f-8i** are determined by the chiral HPLC.

The dr of **8b** (dr: 2.5:1), **8f** (dr: 1:1), **8g** (dr: 1:1), **8h** (dr: 1:1), **8i** (dr: 1:1) have been added.

Compound **8b** is a single diastereoisomer isolated, and the optical rotations have been added.

The previous compound **8d** has been deleted because we have no suitable method to determine the diastereoselectivity.

Please see our revised manuscript and SI.

10. Page 8, line 3: “thus implying” instead of “thus implied”.

Response: Thank you for pointing it out. The “thus implied” has been changed to “thus implying”, please see our revised manuscript.

11. Figures 3 and 4, on pages 8 and 9, should be schemes, not figures.

Response: Thank you for pointing it out. Figures 3 and 4 have been changed to Scheme 5 and 6, please see our revised manuscript.

12. Page 9, Figure 4: many things are missing in the synthetic schemes. What is the relative stereochemistry of compound **9**? It is represented as syn in the supporting info. oaminoaniline, sulfamoyl chloride and choloroacetyl chloride are missing as reagents in steps c, d and e, respectively. The dr for compound **20** should be given.

Response: Thank you for pointing it out.

Compound **9** is anti one.

The missing reagents have been added.

The dr for compound **20** has been added.

Please see our revised manuscript.

13. In the reference section, many things should be corrected, including: scientific names on references 1, 2, 5 and 84 should be written accordingly (italics and first name in capital letters); many proper names throughout the section should begin with capital letters such as Ugi, Passerini, Heyns, Pudovik, Bronsted, Lewis, Organic Synthesis, Schiff, Smiles, Suzuki, Mannich, Petasis, etc. Ga and Yb as well. Ref. 13: BF₃.OEt₂; Ref. 26: RuO₄; Ref. 55: remove 1,2 after “mechanism”; Ref. 73: C(sp³) -C(sp²) instead of c(sp³)-c(sp²).

Response: Thank you for pointing it out. Above errors have been corrected, please see our revised manuscript.

14. Supporting information:

14.1. General information: insert information about the source of commercial reagents.

Response: Thank you for pointing it out. The sources of commercial reagents have been added, and the most starting materials were purchased from Energy Chemical, Bidepharm and Adamas. Please see our revised manuscript.

14.2. General procedures A and B: line 1, “0.36 mmol” instead of “0.38 mmol”; “0.20 mmol” and “0.30 mmol” instead of “0.2 mmol” and “0.3 mmol” (the same for all

compounds **4** in the following pages); “were added” instead of “was added” (same for general procedure C); the ratio of solvents used in the chromatographic purifications should be added (same for general procedure C).

Response: Thank you for pointing it out. Above errors have been corrected, please see our revised SI.

14.3. Page 2, from compound **4b** on: multiplets are represented by a range within which the dashes should be uniform (use either – or -). This lack of uniformity can be seen in the whole experimental section.

Response: Thank you for pointing it out. All ‘-’ have been changed to ‘-’, please see our revised SI.

14.4. Page 3: compound **4g** was prepared from decanal and not from phenylpropyl aldehyde.

Response: Thank you for pointing it out. The “phenylpropyl aldehyde” has been changed to “decanal”, please see our revised SI.

14.5. Page 4, compound **4i**: “(3R,5R,7R)” instead of “(3r,5r,7r)”.

Response: Thank you for pointing it out. The “(3r,5r,7r)” has been deleted because this compound does not involve chirality, please see our revised SI.

14.6. Page 6: state the stereochemistry of **4n**.

Response: Thank you for pointing it out. The stereochemistry of **4n** has been stated. Please see our revised manuscript and SI.

14.7. Pages 11 and 12: compounds **5a** and **5b** are switched (see table 3). Rotate all compounds 5 180° such that they keep the standard used in other parts of the manuscript with the blue portion to the right (same in the spectra).

Response: Thank you for pointing it out. Compounds **5a** and **5b** have been switched. The molecular structure has been rotated, please see our revised SI.

14.8. Pages 12-18, compounds **5c-5x**: put the correct boronic acid for each case and correct the amounts used and the yields obtained according to table 3. (4-Methoxyphenyl)boronic acid was wrongly put in all cases.

Response: Thank you for pointing it out. Above errors have been corrected, please see our revised SI.

14.9. Page 13, **5e**: wrong value for HRMS found.

Response: Thank you for pointing it out. These errors have been corrected, please see our revised SI.

14.10. Page 21, **8b**: melting point and optical rotation are missing.

Response: Thank you for pointing it out. The melting point and optical rotation of **8b** have been added, please see our revised SI.

14.11. Page 22: rotate **8f** by 180°.

Response: Thank you for pointing it out. Compounds **8f** has been rotated 180°, please see our revised SI.

14.12. Page 24, **8i**: melting point is missing; rotate **8k** by 180°.

Response: Thank you for pointing it out. Melting point of **8i** has been added. Compounds **8j** (previous **8k**) has been rotated 180°, please see our revised SI.

14.13. Page 25, line 3: “were added” instead of “was added”; lines 4-5: “Upon completion, the reaction was diluted with CH₂Cl₂ (volume?) and the organic phases were washed with brine (volume?) and dried over Na₂SO₄.” The ratio of eluents should be given for the column chromatography (same for 2nd paragraph).

Response: Thank you for pointing it out. The “was added” has been changed to “were added”; the volume of CH₂Cl₂ and Brine has been added; the ratio of eluents has been added. Please see our revised SI.

14.14. Page 25, 2nd paragraph: “cooled saturated NaHCO₃” (include the temperature and volume); the melting point of the product should be added; compound **9**: state in its name that the compound is racemic.

Response: Thank you for pointing it out. The temperature and volume have been added; the melting point has been added; compound **9** is racemic. Please see our revised SI.

14.15. Include the volume of extraction solvents and aqueous solutions used to wash the organic phases for all compounds from now on. Additionally, the ratio of eluents should be given for the column chromatography in each case. For solid compounds such as 11, 12, 15, 19, 21 and 22, the melting points are missing (compare with literature data when appropriate).

Response: Thank you for pointing it out. The volume of extraction solvents and aqueous solutions has been added.

The ratio of eluents for the column chromatography in each case has been added.

The melting points have been added.

Please see our revised SI.

14.16. Page 27: place the chloroacetyl chloride in the first step of the scheme.

Response: Thank you for pointing it out. The chloroacetyl chloride has been added.

Please see our revised SI.

14.17. Page 29, line 5: “were added” instead of “was added”; “The mixture was cooled to (include the temperature)”; line 6: “overnight” instead of “for overnight”; line 7: “Upon completion, the reaction was diluted” instead of “Upon completion of the reaction, diluted”.

Response: Thank you for pointing it out. Above errors have been corrected, please see our revised SI.

14.18. Page 31, NMR spectra: include captions for all spectra.

Response: Thank you for pointing it out. The captions of all NMR spectra have been added. Please see our revised SI.

Reviewer #3 (Remarks to the Author):

This manuscript reports a novel synthesis of α -hydroxyketones through a three components reaction which can be seen as a combination of Paserini and Petasis reactions. Although both classes of reactions have been extensively studied for many years, to the best of my knowledge, the multicomponent reactions of carbonyls, isocyanides and boronic acids that is described in this paper have not been previously reported. The methodology reported here is fairly general, opening the door to the synthesis of a very wide variety of the synthetically valuable α -hydroxy ketones, in a straightforward manner, from readily available starting materials and in an operationally simple and metal-free procedure.

The scope of the reaction has been studied in detail, as illustrated by the large number of examples included. Additionally, the methodology has been applied to a number of challenging substrates, such as modified drugs and natural products.

I consider that this methodology will be very useful in the context of organic synthesis, and for this reason I think it reaches the high standards to be published in Nature Communications.

The supplementary information is complete and the compounds are well characterized.

Response: Thank you for your favorable comments on our work, we really appreciate it.

Some minor points:

- The reaction with aryl boronic acids is restricted to electron-rich or electron-neutral systems. Nothing is said about electron-poor derivatives, such as the halobenzene boronic acids. Some commentary should be included.

Response: Thank you for pointing it out. We have tried electron-deficient substituted

boronic acids, such as the halophenyl boronic acids, only trace amounts of products could be obtained. And the commentary has been added, please see our revised manuscript.

- It is curious that the reactions with the aldehyde derived from ibuprofene and ketoprofene give rise a to mixture of two diastereoisomes, but the reaction with the analogous naproxene derivative (which is structurally very similar) provides only one isomer. Does the reaction crude present only one stereoisomer in this case? And how was the stereochemistry proposed determined?

Response: Thank you for pointing it out, this is our negligence.

The dr of naproxene derivative is 2.5:1.

Compounds **8b** and **8d** are single diastereoisome isolated.

The dr values of **8f-8i** are determined by the chiral HPLC.

The dr of **8b** (dr: 2.5:1), **8f** (dr: 1:1), **8g** (dr: 1:1), **8h** (dr: 1:1), **8i** (dr: 1:1) have been added.

Compounds **8b** is single diastereoisomer isolated, and the optical rotations have been added.

The previous compound **8d** has been deleted because we have no suitable method to determine the diastereoselectivity.

Please see our revised manuscript and SI.

- In figure 4, A, there are some reagent missing for the synthesis of 11 and 12.

Response: Thank you for pointing it out. The missing reagents have been added, please see our revised manuscript.

REVIEWERS' COMMENTS

Reviewer #1 (Remarks to the Author):

I greatly appreciate the precise revised manuscript which has been modified according to the reviewers suggestions. Therefore now I recommend its publication on Nature Communications.

Reviewer #2 (Remarks to the Author):

The authors have resubmitted their manuscript answering accordingly the questions raised by the referees. I consider the questions to all referees have been properly addressed and all the suggested corrections were done. The quality of the manuscript is a lot better now and in my opinion it is ready to be published but I still suggest some corrections as outlined below:

1. Page 8, line 5: "not be generated" instead of "not generated".
2. Page 12, ref. 1: the scientific name should be in italics.
3. Page 17, ref. 84: Petasis in capital letter.
4. SI, page 2, line 31: "most starting materials" instead of "the most starting materials".
5. SI, page 25, line 826: "diastereomer" instead of "diastereome".
6. In the spectra captions, the solvent and NMR frequency should be stated.
7. One comment: for compounds 8f-i, it was not really necessary to use a chiral HPLC column to determine the dr ratio once the isomers are diastereomers and not enantiomers. A normal HPLC column would suffice.

Carlos Kleber Z. Andrade

Reviewer #3 (Remarks to the Author):

The authors have addressed successfully all my questions and commentaries in this revised version. Therefore, I think that now it is suitable to be published in Nature Communications.

REVIEWER COMMENTS

Reviewer #1 (Remarks to the Author):

I greatly appreciate the precise revised manuscript which has been modified according to the reviewers suggestions. Therefore now I recommend its publication on Nature Communications.

Response: Thank you for your favorable comments on our revised manuscript, we really appreciate it.

Reviewer #2 (Remarks to the Author):

The authors have resubmitted their manuscript answering accordingly the questions raised by the referees. I consider the questions to all referees have been properly addressed and all the suggested corrections were done. The quality of the manuscript is a lot better now and in my opinion it is ready to be published but I still suggest some corrections as outlined below:

1. Page 8, line 5: "not be generated" instead of "not generated".

Response: Thank you for pointing it out. The “not generated” has been changed to “not be generated”, please see our revised manuscript.

2. Page 12, ref. 1: the scientific name should be in italics.

Response: Thank you for pointing it out. The scientific name has been changed to italics, please see our revised manuscript.

3. Page 17, ref. 84: Petasis in capital letter.

Response: Thank you for pointing it out. The “petasis” has been changed to “Petasis”, please see our revised manuscript.

4. SI, page 2, line 31: "most starting materials" instead of "the most starting

materials".

Response: Thank you for pointing it out. The “the most starting materials” has been changed to “most starting materials”, please see our revised SI.

5. SI, page 25, line 826: "diastereomer" instead of "diastereome".

Response: Thank you for pointing it out. The “diastereome” has been changed to “diastereomer”, please see our revised SI.

6. In the spectra captions, the solvent and NMR frequency should be stated.

Response: Thank you for pointing it out. The solvent and NMR frequency has been added, please see our revised SI.

7. One comment: for compounds **8f-i**, it was not really necessary to use a chiral HPLC column to determine the dr ratio once the isomers are diastereomers and not enantiomers. A normal HPLC column would suffice.

Response: Thank you for your comments. Currently we don't have normal HPLC column in our laboratory yet chiral ones available, therefore we used a chiral HPLC column to determine the dr ratio of compounds **8f-i**.

Reviewer #3 (Remarks to the Author):

The authors have addressed successfully all my questions and commentaries in this revised version. Therefore, i think that now it is suitable to be published in Nature Communications.

Response: Thank you for your favorable comments on our revised manuscript, we really appreciate it.